# Blinatumomab Prior to CAR-T Cell Therapy—A Treatment Option Worth Consideration for High Disease Burden

**DOI:** 10.3390/biomedicines10112915

**Published:** 2022-11-13

**Authors:** Paweł Marschollek, Karolina Liszka, Monika Mielcarek-Siedziuk, Blanka Rybka, Renata Ryczan-Krawczyk, Anna Panasiuk, Igor Olejnik, Jowita Frączkiewicz, Iwona Dachowska-Kałwak, Agnieszka Mizia-Malarz, Tomasz Szczepański, Wojciech Młynarski, Jan Styczyński, Katarzyna Drabko, Grażyna Karolczyk, Ewa Gorczyńska, Jan Maciej Zaucha, Krzysztof Kałwak

**Affiliations:** 1Department of Pediatric Bone Marrow Transplantation, Oncology, and Hematology, Wroclaw Medical University, 50-556 Wroclaw, Poland; 2Department of Pediatrics, Medical University of Silesia, 40-758 Katowice, Poland; 3Department of Pediatric Hematology and Oncology, Zabrze, Medical University of Silesia, 41-800 Katowice, Poland; 4Department of Pediatrics, Oncology & Hematology, Medical University of Lodz, 91-738 Lodz, Poland; 5Department of Pediatric Hematology and Oncology, Collegium Medicum, Nicolaus Copernicus University Torun, 85-094 Bydgoszcz, Poland; 6Department of Pediatric Hematology, Oncology and Transplantology, Medical University of Lublin, 20-093 Lublin, Poland; 7Department of Pediatric Hematology and Oncology, Regional Polyclinic Hospital in Kielce, 25-736 Kielce, Poland; 8Department of Hematology and Transplantology, Medical University of Gdansk, 80-214 Gdansk, Poland

**Keywords:** blinatumomab, children, BCP-ALL, CAR-T cell therapy, bridging therapy, hematopoietic stem cell transplantation, tisagenlecleucel

## Abstract

The optimal bridging therapy before CAR-T cell infusion in pediatric relapsed or refractory B-cell precursor acute lymphoblastic leukemia (r/r BCP-ALL) still remains an open question. The administration of blinatumomab prior to CAR-T therapy is controversial since a potential loss of CD19+ target cells may negatively impact the activation, persistence, and, as a consequence, the efficacy of subsequently used CAR-T cells. Here, we report a single-center experience in seven children with chemorefractory BCP-ALL treated with blinatumomab before CAR-T cell therapy either to reduce disease burden before apheresis (six patients) or as a bridging therapy (two patients). All patients responded to blinatumomab except one. At the time of CAR-T cell infusion, all patients were in cytological complete remission (CR). Four patients had low positive PCR-MRD, and the remaining three were MRD-negative. All patients remained in CR at day +28 after CAR-T infusion, and six out of seven patients were MRD-negative. With a median follow-up of 497 days, four patients remain in CR and MRD-negative. Three children relapsed with CD19 negative disease: two of them died, and one, who previously did not respond to blinatumomab, was successfully rescued by stem cell transplant. To conclude, blinatumomab can effectively lower disease burden with fewer side effects than standard chemotherapeutics. Therefore, it may be a valid option for patients with high-disease burden prior to CAR-T cell therapy without clear evidence of compromising efficacy; however, further investigations are necessary.

## 1. Introduction

The outcomes of relapsed or refractory (r/r) B-cell precursor acute lymphoblastic leukemia (BCP-ALL) treatment have improved significantly with the application of immunotherapy [1,2,3,4,5]. Clinically available drugs are directed towards surface proteins expressed on B-lineage cells, named cluster of differentiation (CD). CD19 is the most exploited target antigen for therapies engaging T cells: both for bispecific antibody blinatumomab and commercially available chimeric antigen receptor T-lymphocytes, tisagenlecleucel (CAR-T) [1].

CAR-T cell technology is an example of cell gene therapy, personalized to target the unique type of an individual patient’s disease. CAR-T cells are the patient’s own T lymphocytes obtained after ex vivo genetic manipulation. Gene encoding chimeric antigen receptor (CAR) anti-CD19 is introduced into the genome of T cells with the help of a lentiviral vector. When infused into the bloodstream, CAR-T cells are able to penetrate tissues, self-activate, amplify, eliminate neoplastic cells and act long-term. CD19-targeted CAR-T cell therapy has become an efficacious option for the treatment of r/r BCP-ALL [4,5,6]. Clinical reports on the use of blinatumomab in relapsed and refractory BCP-ALL show moderate efficacy. For example, a phase I/II study by Stackelberg et al. found that among 70 patients included in the trial, 27 (39%) achieved complete remission, 14 of whom (52%) achieved complete minimal residual disease response [2].

As both blinatumomab and CD19 CAR-T are targeting the same CD19 antigen displayed on the surface of BCP-ALL cells, it is generally believed that these two therapies exclude each other, but recent data may challenge this statement. The role of bridging therapy is to prevent rapid leukemia progression. Optimally, it should be effective, leading to low disease burden or even negative MRD without significant toxicity [7]. In this aspect, blinatumomab plays the role satisfactorily [2,3]. Although a low disease burden (below 5% of blasts in bone marrow) is related to better final clinical outcomes [7,8], bridging therapy targeting CD19 may promote the development of CD19-negative leukemic cells. The place for blinatumomab in the treating regimen in children with r/r leukemia has been widely discussed recently [1,6,7,8,9]. Blinatumomab can be included in treatment at different time points, depending on the patient’s clinical situation. In relation to CAR-T cell therapy, it can be used before apheresis, as a standard treatment line, or as a bridging therapy to control disease burden while manufacturing CAR-T cells [1,6,7]; however, according to many experts, its use remains very controversial (A. Balduzzi, personal communication).

Here, we report our clinical observations of the impact of prior blinatumomab treatment on the outcome of subsequent CD19 CAR-T cell therapy in seven pediatric patients with r/r BCP-ALL treated at Wroclaw Medical University between January 2021 and March 2022.

## 2. Materials and Methods

Seven patients with r/r BCP ALL treated with CAR-T cells (tisagenlecleucel (Kymriah^®^, Novartis) in the Department of Pediatric Bone Marrow Transplantation, Oncology, and Hematology of Wroclaw Medical University are retrospectively assessed. With the exception of patient #3, all patients received at least one course of blinatumomab at a daily dose of 15 µg/m^2^ not earlier than 6 months before apheresis. Blinatumomab was administered due to disease refractoriness to standard chemotherapy and/or as a bridging therapy to achieve lower MRD status. A complete course of blinatumomab treatment was defined as a 28-day continuous infusion. Evaluation of response to blinatumomab was based on MRD in accordance with Myers et al. criteria [7]. The lymphodepleting regimen in all of the patients was: fludarabine 4 × 30 mg/m^2^ (days −7 to −4 before CAR-T infusion) and cyclophosphamide 2 × 500 mg/m^2^ (days −7 to −6 before CAR-T infusion). Cytokine release syndrome (CRS) was diagnosed, and its intensity was assessed according to the Penn grading scale [10]. Neurotoxicity was evaluated with the ASTCT Consensus Grading of ICANS [11]. The first assessment of response to CAR-T cell therapy was performed on day +28 after the infusion; then, the observation was adjusted for each individual patient. Complete remission (CR) was defined as M1 bone marrow (<5% blasts in cytomorphology) without evidence of extramedullary disease. MRD-PCR analysis was performed during every bone marrow assessment (based on the detection of clone-specific immunoglobulin (Ig) and T-cell receptor (TCR) rearrangements; sensitivity of 10^−5^). Relapse was defined as an MRD positivity in previously negative patients. CD19 status was assessed with flow cytometry. Loss of B-cell aplasia (BCA) was defined as a reappearance of >5 CD19+ cells/µL in peripheral blood. A commercial flow cytometry panel (Miltenyi Biotec, Germany) was used for CAR-T cell detection [12]. At first, CD-19 CAR-T were specifically bound with a biotinylated CD19 antigen (CD19 CAR Detection Reagent, anti-human, Biotin, REAfinity), and then, the labeled CAR-T cells were stained with a fluorochrome-conjugated anti-biotin antibody (anti-biotin, REAfinity). The control tube was incubated with the same antibody mix without the Detection Reagent. Finally, cells were assessed on a CANTO (BD Bioscience, Franklin Lakes, NJ, USA) flow cytometer. The study was conducted according to the principles of the Declaration of Helsinki (version 7, October 2013).

## 3. Results

Six patients received blinatumomab with a median time of 13.5 days between the last dose and apheresis. Two patients (#2 and #3) were administered blinatumomab as a bridging therapy, and one of them (#2) received it before apheresis and also as a bridging therapy. Table 1 presents detailed information on the rest of the patients’ conditions and administered treatment.

All patients received at least one cycle of blinatumomab, except patient #5, whose treatment was terminated due to disease progression assessed on day 14 [13]. Patient #7 received blinatumomab for the treatment of a previous relapse just before the hematopoietic stem cell transplantation (HSCT) (101 days before the CAR-T apheresis) and was therefore also included in this study. In our analysis, we observed only one case of blinatumomab non-response (#1).

The median age at CAR-T cell infusion was 7.5 years. Regarding indications for the therapy: 28.6% (2/7) of children had primary refractory disease, 28.6% (2/7) had secondary resistance (a refractory relapse), and 42.9% (3/7) suffered from relapse after HSCT. At the time of the CAR-T infusion, three patients had negative MRD, and all patients were in CR. The median time from apheresis to infusion was 55 days (range 39–67 days). Table 2 presents more details concerning apheresis and MRD status before CAR-T cell infusion, as well as information on the percentage of CD19 expression on leukemic cells at the time of diagnosis of r/r disease.

In the first days after CAR-T infusion, in six out of seven patients (all, except patient #1, 85.7%), cytokine release syndrome was observed: in five patients, stage I; in one child (patient #2), stage II (requiring the use of three doses of tocilizumab). One of them (patient #5) was also diagnosed with stage I neurotoxicity. 

All patients were in CR at day +28 post CAR-T cells infusion: six out of seven were MRD negative, and one child had PCR-MRD positive <10^−4^ (Patient #5). B-cell aplasia (BCA) was recorded in all patients after tisagenlecleucel infusion.

Currently, among survivors, the median follow-up time is 509 days (range 291–653 days). Four patients (57.1%) remain in complete remission, with negative PCR-MRD and B-cell aplasia (median observation 497 days, range 291–621 days). In three children (#1, #5, and #7; 42.9%), based on PCR and flow-cytometry-MRD re-emergence, CD19-negative relapses were diagnosed. Figure 1 illustrates the clinical outcomes of patients who relapsed after CAR-T cell infusion. Patient #1 is again in complete remission now, 13 months after the salvage matched unrelated donor HSCT. BCA was observed even up to 4 months after the relapse (277 days from the CAR-T infusion). Patient #5 died 5 months after CAR-T cells infusion due to disease progression. The loss of BCA was not observed in this patient. In patient #7, who died 4.5 months after the infusion, the loss of BCA was observed 4 days before the diagnostic bone marrow biopsy (57 days from the CAR-T cells infusion), confirming CD19-negative relapse. Neither salvage inotuzumab ozogamycin block nor rescue chemotherapy were effective (both BM and extramedullary lesions were confirmed).

Table 3 presents the number of CAR-T cells in peripheral blood assessed in patients at certain time points.

## 4. Discussion

Treatment of r/r BCP-ALL has significantly improved thanks to novel immunotherapeutic drugs. Nevertheless, the administration of these new agents brought questions about their possible impact on leukemic cells’ immunological features and their further clinical implications [8,14]. Blinatumomab induces remission in relapsed or chemorefractory BCP-ALL (also in Philadelphia-positive leukemia) [4,5,15,16], but it is associated with CD19 antigen modulation [1,14,17]. The available data regarding the impact of prior exposure to blinatumomab on subsequent CAR-T cell therapy are limited [7,8]. A special concern is put on CD19 antigen loss and downregulation, more common after CD19-targeted treatment, which was discussed in detail by Mejstríková et al. [18]. Additionally, the study by Pillai et al. revealed that in patients treated with CAR-T cells with prior exposure to blinatumomab, the rate of MRD-negative remissions was significantly lower. There was also a higher incidence of CD19-negative MRD or relapse [19]. On the other hand, the clinical study by Myers et al. shows comparable results of CAR-T cell therapy in Blina-naive and Blina-responders (CR in 93.5% and 92.9% of patients, respectively), with inferior outcomes in the Blina-non responders’ group, similarly to those with high-disease burden [7]. It may indicate that possible CD19 antigen escape is not the primary reason for therapy failure. Analysis of our center’s clinical outcomes, as well as the study by Myers et al., confirms the efficacy of blinatumomab administration both before apheresis and as a bridging therapy. We used blinatumomab as a salvage therapy in children with chemorefractory disease with the aim of achieving low disease burden or MRD negativity prior to CAR-T cell therapy. Blinatumomab proved to be effective in the majority of patients; only one child was classified as a blinatumomab non-responder.

At the time of CAR-T cell infusion, all patients were in complete remission (CR), four of whom had low positive PCR-MRD, and three were MRD-negative. These results confirm blinatumomab efficacy in achieving optimal disease status before CAR-T cell infusion. Therefore, we may speculate that blinatumomab administration prior to CAR-T cell therapy effectively clears the high-disease burden, still sparing a few potential target cells, which will promote later CAR-T cell activation and persistence. The median time between blinatumomab exposure and CAR-T cell therapy was significantly shorter compared with the Myers study (64 vs. 131 days), and this does not seem to negatively impact CAR-T efficacy.

On day +28, six out of seven patients had negative MRD. This is consistent with the tisagenlecleucel registration trial, where 95% of patients with CR were MRD (-) at this time point [4]. It is worth mentioning that the ELIANA study excluded patients previously treated with other B-cell targeted therapies (i.e., blinatumomab).

Four out of seven children have been followed up for at least one year. Three of them remain in CR with negative MRD. In large cohorts presented in ELIANA [4] and CIBMTR [5] studies, the 1-year event-free survivals were 57.2% and 52.4%, respectively.

The impact of blinatumomab therapy as a bridge to CAR-T cells infusion was especially relevant in patient #2, in whom the above-mentioned immunotherapy constituted the only way to achieve low-disease burden before tisagenlecleucel administration. Despite previous anti-CD19 targeted treatment, he developed CRS grade II with high levels of IL-6 and ferritin, so as persistently high CAR-T cell number in peripheral blood. It shows that there still was a meaningful amount of target cells for CAR-T activation, proliferation, and persistence. It is clear that without blinatumomab, the chance of curing this patient would have been extremely poor.

The majority of patients respond to anti-CD19 immunotherapy. Short-term response rates are remarkably high; however, a significant group of patients relapse anyway [4,5,13]. Both CD19-positive and negative relapses occur—in our study, all of them were CD19-negative and occurred in children who had significantly weaker CD19 expression in leukemic cells at the diagnosis of r/r disease. In the ELIANA trial, CD19 CAR-T therapy results were similar—fifteen patients had CD19- relapse (despite no prior administration of blinatumomab), six patients had unknown CD19 status, and only one had CD19+ relapse [4]. CD19-positive relapses are generally associated with poor T cell function or early CAR-T cell disappearance, and its prevalence is more common in late relapses [7]. CD19-negative relapses represent a new mechanism of leukemic blast escape [20,21]. There is a hypothesis that these escape strategies (i.e., mutations affecting the CD19 gene or CD19 splicing variants) can be driven by the use of CD19-targeted immunotherapeutic agents [14,17]. Other studies suggest that minimal CD19-negative subpopulations can pre-exist in the leukemic blasts at diagnosis but are selected and start to dominate after CD19 CAR-T cell therapy [22,23], and we might speculate that our patients developed a relapse in the a.m. mechanism. Patients with CD19-negative relapsed leukemia have a very poor prognosis [14]. Nevertheless, the prognosis of patients qualified for immunotherapy is a priori unfavorable.

In our opinion, blinatumomab administration prior to CAR-T cell therapy can be beneficial in many aspects. Only two potential drawbacks should be mentioned: an unknown risk of CD-19 negative selection of leukemic cells and a potential loss of target cells for CD19 CAR-T cell activation and persistence. The advantages of blinatumomab administration before CAR-T cell therapy include non-toxic maintenance therapy enabling achievement of optimal peripheral blood counts for successful apheresis (blinatumomab does not impact T-cell number and their function); potential efficacy in achieving low disease burden or even negative MRD prior to CAR-T cell therapy (which will further improve the outcome by better control of MRD and by decreasing the risk of severe CRS/neurotoxicity after CAR-T cell infusion).

Innovative strategies to avoid antigen loss include allogeneic hematopoietic stem cell transplantation after CAR-T cell therapy [14]. The study of Shah et al. showed that patients with r/r leukemia may benefit from the sequential therapy with CD19 CAR-T cell followed by allo-HSCT [24]. In our opinion, patients with weaker CD19 expression in blast cells should be given CAR-T cell therapy as a bridge to HSCT. The drawback is that many patients have already received HSCT before anti-CD19 immunotherapy and are not eligible for the next transplantation due to its high toxicity.

The novel multiple targeted (CD22, CD20) immunotherapeutic agents [14] can also become a solution similarly to the multidrug chemotherapeutic regimens that improve clinical outcomes significantly. A recent report by Libert et al. reveals that in patients with CD19 loss after targeted immunotherapy, there was a corresponding decrease in CD22 expression, indicating that CD19/CD22 CAR-T may not necessarily be a good strategy [21]. On the other hand, the optimistic message comes from a recent report by Ceolin et al., showing no impact of treatment with inotuzumab ozogamycin on CAR-T cell therapy in children with r/r ALL [25].

## 5. Conclusions

In our clinical experience reported here, the advantages of blinatumomab as pre-CAR-T therapy clearly outweigh its drawbacks. Blinatumomab administration prior to CAR-T cell therapy seems to be a safe and beneficial option for providing non-toxic maintenance therapy in patients after intensive and aggressive treatment. It might lead to achieving a balance between preventing leukemia progression and ensuring optimal circumstances for apheresis and further immunotherapy. Achieving low MRD status before lymphodepleting therapy remains challenging, and blinatumomab can be a good option for treating patients with high-disease burden or those with the chemorefractory disease. In our observation, blinatumomab proved to be efficient in decreasing MRD levels without compromising the patient’s clinical condition or subsequent immunotherapy; however, further investigations, including larger randomized studies, are needed to fully assess the efficacy of blinatumomab as pre-CAR-T therapy. There is a lack of pre-clinical studies, which would be helpful to better understand the specific effects of combining BiTE and CAR-T therapy.

## Figures and Tables

**Figure 1 biomedicines-10-02915-f001:**
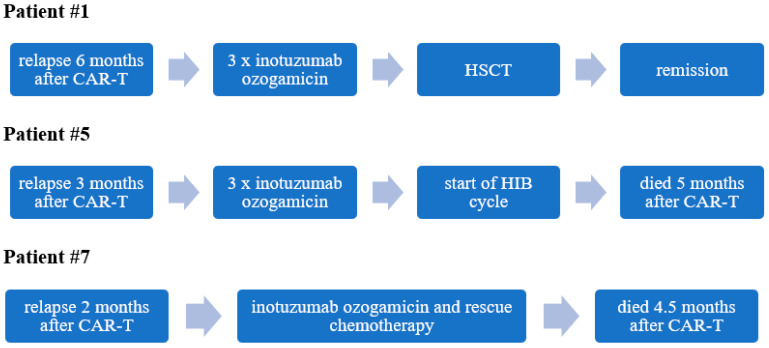
Clinical outcomes in relapsed patients; abbreviations: HIB—chemotherapy cycle according to IntReALL 2010 Protocol; HSCT—hematopoietic stem-cell transplantation.

**Table 1 biomedicines-10-02915-t001:** Disease and treatment characteristics; * age at the time of CAR-T cells infusion; abbreviations: Blina—blinatumomab, CR—complete remission, Dexa—dexamethasone, Ino—inotuzumab ozogamicin, MRD—minimal residual disease; 6-MP—6-mercaptopurine, VCR—vincristine, HSCT—hematopoietic stem cell transplantation, Prot. I—Protocol I, Cons. A—Consolidation A, Cons. A + B ext.—Consolidation A + B extended according to AIEOP-BFM 2017 POLAND, HIB, HC 1 and 2, SIA—chemotherapy cycles according to IntReALL 2010 Protocol; ** according to Myers et al. JCO 2022 [7].

Patient No# (Age *, Sex)	Genetic Lesions	Indication for CAR-T Therapy	Treatment before Apheresis	Bridging Therapy	Blina Response **	Follow-Up (Days)/ Outcome
#1 (7.5 years, girl)	t(17;19) TCF3::HLF	Primary refractory disease	Prot. I, Cons A + B_ext_, HR1, 1× Blina	HR2 + Venetoclax, HR-3 + Venetoclax	No response	653/ relapse
#2 (5.5 years, boy)	t(12;21) ETV6::RUNX1	2nd relapse, after 1st HSCT	HIB, 3× Ino, 1× Blina	1× Blina	MRD-positive CR	621/ CR
#3 (15.5 years, boy)	none	3rd relapse, after 1st HSCT	None—apheresis on relapse occurrence	2× VCR + Dexa, 1× Blina	MRD-negative CR	509/ CR
#4 (1.5 years, girl)	t(11;19) KMT2A:: MLLT1	Primary refractory disease	Prot. I, Cons A, 2× Blina	HIB	MRD-positive CR (unstable with increasing disease)	485/ CR
#5 (7.5 years, girl)	complex caryotype with hipodiploidy	Secondary refractory disease	HIB, HC1, HC2, 1× Blina (20 days)	3× Ino	MRD-positive CR (with increasing disease on day 14 of blina cycle)	148/ relapse, death
#6 (7.5 years, boy)	ETV6::RUNX1(12;21) with additional gene fusion ETV6::RUNX1(12;21)	Secondary refractory disease	SIA, 2× Blina	6-MP + VCR	MRD-negative CR	291/ CR
#7 (5.5 years, girl)	none	2nd relapse, after 1st HSCT	HIB, 2× Blina, alloHSCT	3× Ino	MRD-positive CR	143/ relapse, death

**Table 2 biomedicines-10-02915-t002:** Details of apheresis and CAR-T infusion; abbreviations: Blina—blinatumomab; CD3+—CD3+ lymphocytes; CD19+–CD19+ lymphocytes; n/a—not assessed, neg.—negative; WBC—white blood cells, PB—peripheral blood, r/r—relapsed or refractory disease.

Patient (#)	CD19 on Blast Cells at the Time of Diag. r/r Disease (%)	Blina-Apheresis Interval (Days)	WBC (Apheresis; Cells × 10^3^/µL)	CD3+ in PB (Apheresis; Cells/µL)	CD19+ in PB (Apheresis; Cells/µL)	BLINA- CAR-T Interval (Days)	Apheresis- CAR-T Interval (Days)	CD19+ in PB before CAR-T (Cells/µL)	MRD-PCR before CAR-T
#1	40–50	16	2.09	588	0	-	66	0	neg.
#2	100	8	2.94	1011	0	11	48	0	6 × 0 10^−3^
#3	100	-	7.3	2192	158	28	67	3	neg.
#4	97–98	14	4.12	1132	151	-	57	2	2 × 10^−2^
#5	60–70 (dim)	9	3.52	1132	0	-	55	0	<10^−4^
#6	100	13	2.35	432	n/a	-	39	1	neg.
#7	68	101	3.78	648	135	-	44	0	<10^−4^

**Table 3 biomedicines-10-02915-t003:** CAR-T expansion in the patients’ blood; abbreviations: n/a—not assessed.

Patient (#)	CAR-T in Peripheral Blood (cells/µL)						
	Day 7	Day 14	Day 28	2 Months	3 Months	6 Months	Present
1.	20	29	7	3	1.9	2	-
2.	71	343	134	53	543	38	11
3.	28	5	1	2	1	2	1.6
4.	23	26	2	n/a	2	3	1
5.	7	3	9	17	-	-	-
6.	61	8.2	10	n/a	5	2	2
7.	12.7	5.5	2.5	9	2	-	-

## Data Availability

Not available.

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
