# Peer review of "Blinatumomab Prior to CAR-T Cell Therapy—A Treatment Option Worth Consideration for High Disease Burden"

_biomedicines, 2022, doi:10.3390/biomedicines10112915_

Round 1

Reviewer 1 Report

In the manuscript titled "Blinatumomab prior to CAR-T cell therapy should not be disregarded since it might be optimal in selected patients with refractory acute lymphoblastic leukemia", Marschollek et all report on their clinical experience with the combination of Blinatumomab (Blina) and CD19 CAR-T at the Wroclaw Medical University. Authors have assembled their observations with patients treated with Blina immediately prior to CAR-T apharesis and/or using Blina as bridging/conditioning therapy prior to CAR-T infusion. 

The clinical observations reported by Marschollek et al are very interesting and worthy of scientific report. The negative dogma around the combination of Blina and CD19 CAR-T is complex and not necessarily fully supported in the literature, so observational reports such as this are critical to identify clinical paths forward. 

The manuscript itself has some issues with the writing, but all could be addressed by responding to the comments below:

(1) The current title is poorly written. A more plainly written title would be better. Suggestion "Report of Clinical Experience with Blinatumomab Prior to CAR-T in Seven Pediatric Patients"

(2) Abstract sentence "Here, we report a single-center experience in seven children with chemorefractory BCP-ALL treated with blinatumomab before CAR-T cell therapy either to reduce disease burden before apheresis (six patients) or as bridging therapy in two patients with active disease. bridging therapy (two patients).". 

(3) Acronym MUD-HSCT in the abstract is not defined. Using the more general term "stem cell transplant" would be fine for the abstract.

(4) Sentence "To conclude, blinatumomab appear to hamper rapid progression can effectively lower disease burden with less side effects than standard chemotherapeutics and may be a valid option for patients with high-disease burden due to its efficacy in decreasing MRD levels without clear evidence of compromising efficacy of subsequent prior to CAR-T cell therapy, without clear evidence of compromising efficacy

(5) "CD19 is the most exploited target antigen for therapies engaging T cells..."

(6) "The study by Stackelberg et al. confirms moderate efficacy of blinatumomab in children with r/r BCP-ALL. Clinical reports on the use of blinatumomab in relapsed and refractory BCP-ALL show moderate efficacy. For example, a phase I/II study by Stackelberg et al found that Aamong 70 patients..."

(7) In paragraph 2 of the introduction authors should add a sentence to introduce CD19 CAR-T therapy, and provide a reference for the efficacy of CAR-T for treatment of BCP-ALL. 

(8) In paragraph 3 of the introduction, the use of the term "conflict of interest" is not appropriate. Suggested edit: "The ‘conflict of interest’ starts, since CAR-T cells need to attach to the CD19 antigen to attack blasts, self-activate and persist for months or longer. It As both blinatumomab and CD19 CAR-T are targeting the same CD19 antigen displayed on the surface of BCP-ALL cells, it is generally believed that these two therapies exclude each other..."

(9) As it seems this is more of an observational report rather than specific clinical study, it is suggested to edit the last sentence of the introduction. "The main aim of this study was to assess Here we report our clinical observations of the impact of prior blinatumomab treatment on the outcome of subsequent CD19 CAR-T cell therapy in seven pediatric patients with r/r BCP-ALL treated at Wroclaw Medical University Hospital between [date] and [date]. "

(10) "With the exception of patient 3, all patients received at least one course of blinatumomab at a daily dose of 15 µg/m2 not earlier than 6 months before apheresis.

(11) Was the course of blinatumomab initiated or completed "not more than 6 months before apharesis"?

(12) Details should be provided in the materials and methods as to how MRD status was assessed. 

(13) It is not clear whether a full course of blinatumomab was administered for those patients who received Blina as bridging therapy. 

(14) "Finally, cells were assessed on a CANTO (BD Bioscience, USA) flow cytometer.

(15) In table 1: "Prot. I, Cons A+Bext, HR1, I1x Blina"

(16) The information provided on the interval between CAR-T and Blina is very interesting. Authors should comment in the results or discussion section on whether this is a shorter interval between Blina and CAR-T then most other studies/retrospective analyses. 

(17) "In the first days after the CAR-T infusion, in 6/7 patients (all, except patient #1, 85.7%) cytokine release syndrome was observed" (not clear if this is Blina or CAR-T infusion"

(18) Authors do not reference Table 3 in the text. Authors should provide some context for these results if they are going to be included. 

(19) Table numbering should be in numeral format (1,2,3...) , not roman numerals. 

(20) "Majority of  The majority of BCP-ALL patients respond to anti-CD19 immunotherapy."

(21) "In the ELIANA trial of CD19 CAR-T therapy..."

(22) "In our opinion, blinatumomab administration prior to CAR-T cell therapy is can be beneficial..."

(23) "The worrying point, In a recent reported by Libert et al., is that in their study in patients with CD19 loss after targeted immunotherapy, there was a corresponding decrease in CD22 expression [20], indicating that CD19/CD22 CAR-T may not necessarily be a good strategy."

(24) "In our clinical experience reported here, the Aadvantages of blinatumomab as pre-CAR-T therapy clearly outbalance outweigh its drawbacks. "

(25) "Achieving low MRD status before lymphodepleting therapy remains challenging, and Blinatumomab can may be a good adequate option for treating patients with high-disease burden or those with chemorefractory disease."

(26) "however further investigations including larger randomized studies are necessary needed to fully assess the efficacy of blinatumomab as pre-CAR-T therapy". 

(27) Authors should consider adding a mention of the lack of pre-clinical studies to better understand the specific effects of combining BiTE and CAR-T therapy in their concluding statements. 

Overall this is an important report that should be published, though authors must address the critical edits suggested above. Given the importance of the suggested edits, authors should revise and resubmit. 

Reviewer 2 Report

This is a case series of 7 pediatric patients with R/R B cell ALL who were treated with blinatumomab prior to CAR-T. Blinatumomab is frequently used prior to apheresis or to lymphodepletion as bridging but there are concerns that it might reduce the efficacy of subsequent CAR-T therapy. This was mainly shown in the study by Pillai et al (ref 7 in the manuscript) which was referenced but not adequately addressed by the authors. I don't think that this 7 patient case series can answer this important question or add to our knowledge. In addition, the authors seem to indicate that achieving MRD negativity prior to CAR-T is an optimal status for CAR-T success but this also unknown as the registrational trial by Maude et al only included patients with >5% blasts in the marrow. Overall, I found this case series confusing and potentially misleading.

Reviewer 3 Report

Outcomes among paediatric patients and adolescents and young adults (AYAs) with B-cell precursor acute lymphoblastic leukaemia (BCP-ALL) have continuously improved in recent decades, with long-term survival rates now reaching 90% in children and 70% in young adults treated on contemporary protocols. However, 15–20% of paediatric patients and almost 30–40% of young adult patients relapse or remain refractory to primary therapy. Outcomes for patients who experience early bone marrow relapse (<18 months), have ≥2 relapses, a relapse after allogeneic haematopoietic stem cell transplantation (HSCT) or who are refractory to induction therapy are historically very poor. Until recently, the standard of care for these relapsed/refractory (R/R) patients was based on intensive block chemotherapy followed by consolidation with HSCT if deep remission could be achieved.

In the last decade, however, the advent of targeted immunotherapies, e.g., the bispecific antibody blinatumomab (anti-CD19/anti-CD3), the antibody-drug conjugate inotuzumab ozogamicin (anti-CD22) and chimeric antigen receptor (CAR) T-cell therapy has provided novel tools to achieve responses in patients with resistant leukaemia and dramatically augmented treatment options for R/R BCP-ALL. There are limited data regarding the impact of prior blinatumomab exposure on subsequent CD19-CAR outcomes. It was previously shown that the impact of sequential CD19 targeting, blinatumomab nonresponse and high-disease burden were independently associated with worse RFS and EFS, identifying important indicators of long-term outcomes following CD19-CAR. Here in this article authors aim to assess the impact of prior blinatumomab on the outcome of subsequent CD19 CAR T cells therapy in pediatric patients. Although the cohort is small, this article is trying to solve the gaps for the current practice.

1. The title of the article is too long and seems confusing. Please revise it with the point you want to make in the manuscript like ‘Blinotumomab prior to CAR T cell-An Option for high disease burden

2.  It is better not do a definite conclusion as stated in the abstract because this is only a case series including 7 patients, ‘ Its administration prior to CAR T cell therapy does not seem to affect the outcome negatively, however further investigations are necessary’

3. Please erase either the repetitive sentences in the introduction ‘ CD19 is the target antigen for both blinotumomab and commercially available chimeric antigen receptor T-lymphocyes, tisagenlecleucel (CAR-T), while inotuzumab ozogamicin targets CD22.’ and ‘These are bi-specific molecules that attact T cells by binding with CD3 receptor and direct their action selectively against tumour cells due to their affinity to particular cancer antigen. Blinatumomab specifically targets CD19 antigen.’

4. Please revise the sentence. ‘In relation to CAR T cell therapy it is used both before an apheresis or as a bridging therapy’

5. In table 3, what does column CD19+ in PB before CAR T show? Is it cells/ul?

6. In flowchart I, what does HIB stands for?

7. The patients that do not respond to blinatumomab are inherently higher risk patients or part of a CD19- reistant population. I think the authors might consider treating patient 1 a strategy rather than CAR T cells and discuss this issue

8.  What was the lymphodepleting strategy prior to CAR T?

 9. In the ZUMA-3, which evaluated brexu-cel in patients with relapsed/refractory B cell ALL, patients who had received prior blinatumomab were included if their leukemic blasts had >90% CD19 expression. Here in this study, authors can add the percentage of CD19 expression in leukemic blasts. Since patient 2 had no CD19 expressed cell, how did you consider CAR T or blinotumumab bridging?

Round 2

Reviewer 1 Report

Authors have adequately addressed my concerns. The title of the manuscript could be improved further: "Blinatumomab prior to CAR-T cell therapy - a treatment option worth consideration for high disease burden"

Author Response

We greatly appreciate all your comments that have helped us improve our research. We have improved the title according to your suggestion.

Reviewer 2 Report

It appears that the authors have made the appropriate edits

Author Response

We greatly appreciate all your comments that have helped us improve our research.

Reviewer 3 Report

The authors have made the suggested revisions. 

Author Response

(The authors gave the same response as above.)
